# Resveratrol Effects on the Reproductive System in Ovariectomized Rats: Deciphering Possible Mechanisms

**DOI:** 10.3390/molecules27154916

**Published:** 2022-08-02

**Authors:** Ganna Zaychenko, Olena Stryga, Oksana Sinitsyna, Anna Doroshenko, Oksana Sulaieva, Tetyana Falalyeyeva, Nazarii Kobyliak

**Affiliations:** 1Pharmacology Department, Bogomolets National Medical University, 01601 Kyiv, Ukraine; farm26so@ukr.net (O.S.); annadoroshenko2022@ukr.net (A.D.); 2Department of Clinical Pharmacology, Institute of Improvement Qualification of Pharmacy Specialists, National University of Pharmacy, 61002 Kharkiv, Ukraine; sinitsyna.ksu@ukr.net; 3Medical Laboratory CSD, 03122 Kyiv, Ukraine; o.sulaieva@csd.com.ua (O.S.); tetyana.falalyeyeva@knu.ua (T.F.); 4Department of Biomedicine, Taras Shevchenko National University of Kyiv, 01033 Kyiv, Ukraine; 5Endocrinology Department, Bogomolets National Medical University, 01601 Kyiv, Ukraine

**Keywords:** resveratrol, polyphenols, hyaluronic acid, menopause, hypoestrogenic disorders, vaginal gel

## Abstract

Phytoestrogen resveratrol (R) has been demonstrated to benefit human reproductive health. However, R bioavailability and pharmacokinetics are still problematic under oral supplementation. We used an experimental vaginal gel with R and hyaluronic acid (HA) to improve bioavailability and pharmacokinetic properties. The study aimed to assess the impact of vaginal R-HA gel on the reproductive system in ovariectomized rats. Methods: The study was carried out on Wistar female rats. It investigated the body weight, tail temperature, vaginal pH, estrogen and progesterone blood levels, and immunohistochemical biomarkers (COX2, Casp-3, Bcl-2, and VEGF). Animals were divided into control animals; ovariectomized rats (OVX); and OVX group treated with vaginal 0.5% R-HA gel (0.5%, 0.1 mL, daily 28 days). Results: The R-HA gel’s therapeutic effect was manifested by slowing weight gain by 17% (*p* < 0.001), less pronounced symptom of fever at the root of the tail by 9% (*p* < 0.001) and lowering the vaginal pH to 4.4–4.5 compared with OVX rats. The anti-inflammatory effect and the reduction of COX-2 expression in vagina were accompanied by antiapoptotic impact of RA-H on endometrium, associated with the decreased Casp-3 expression (*p* < 0.001) and elevated Bcl-2 score in endometrial glands (*p* = 0.01). Together with enhanced VEGF expression in endometrial glands (*p* < 0.001) and stromal cells (*p* = 0.007), these changes prevented endometrial atrophy (*p* < 0.001) after ovariectomy. Thus, this study substantiates the feasibility of developing an innovative topical drug with R and HA for treating hypoestrogenic disorders.

## 1. Introduction

Menopause is one of the causes of osteoporosis, cardiovascular complications, cognitive impairment, and metabolic disorders in women [1,2,3]. With the onset of menopause in women, there are genetically programmed age-related changes associated with a gradual decline in ovarian function. Premature exclusion of the ovaries, which occurs after surgical or pharmacological castration due to ovarian depletion syndrome after assisted reproductive technologies, leads to abrupt changes in the hormonal regulation in younger women. Different etiologies of menopausal disorders develop against the background of estrogen deficiency and can cause pathological symptoms and menopausal syndrome, which reduces the quality of life of women.

Menopausal symptoms resulting from ovarian failure commonly cause menopausal women to consult their physicians (hot flashes, bone mass loss, urinary complaints, vaginal dryness, and dyspareunia caused by vaginal atrophy) [1,4]. This period is characterized by the loss of hormonal function of the ovaries that dramatically affects the female reproductive system, inducing changes in uterus, vagina, and vulva structure, microbiome, local immunity, and functioning [5]. With increased life expectancy, the impact of vulvovaginal atrophy (VVA) on the quality of life, sexual function, and pelvic floor health is becoming more evident in the current practice of medicine [2,6]. In this hypoestrogenic state, the vaginal epithelium becomes thinner, glycogen production decreases, and, as a result, lactic acid production is reduced, increasing vaginal pH. The change in pH encourages the overgrowth of nonacidophilic coliforms [3].

According to the role of estrogen, the deficiency causing VVA as first-line treatment estrogens are recommended for vaginal atrophy resulting from hypoestrogenism [2,7]. Systemic administration of estrogen as a part of post-menopausal hormone therapy would ameliorate urogenital atrophy. Still, after the increased risk of venous thromboembolism, stroke, and breast cancer, assessed in the Women’s Health Initiative (WHI) trial [8], many women no longer choose such therapy [9,10]. To avoid or reduce possible adverse effects and the risks known from systemic administration of estrogens, vaginal estrogen therapy is generally recommended unless other symptoms of menopause, such as hot flashes, are present [10]. However, there remains some fear that vaginal estrogen treatment may be absorbed in the circulation and increase adverse systemic estrogen-dependent effects [11]. Despite promising evidence, labeling the commercially available vaginally applied estrogens regarding contraindications and warnings is mainly the same as for systemically applied estrogens due to the “class-labeling”, generally often performed by health authorities. The question is if those side effects and risks are dependent on or different by the type of estrogens used vaginally and if the risk of systemic action could be reduced with low-dose or with new “ultra-low dose” estrogen products [10].

Symptoms of vaginal dryness and VVA are progressive and various treatments, including different over-the-counter regimens such as vaginal moisturizers and lubricants, are proper [12]. The survey in the US showed that vaginal dryness could also occur in younger women due to the utilization of antiestrogen medications [13]. Therefore, vaginal lubricant therapy may help improve post-menopausal atrophic vaginitis. It has been shown that vaginal moisturizers and lubricants provide fast relief for vaginal dryness and dyspareunia and may attenuate discomfort for painful intercourse [14,15].

Today, there is a lack of topical drugs that complement systemic effects or eliminate mainly genitourinary manifestations (dryness, itching, irritation of the vaginal mucosa, dyspareunia, urinary incontinence, etc.). A promising way to solve modern pharmacology’s current scientific problem is to develop vaginal drugs with multimodal action with phytoestrogens and hyaluronic acid.

Phytoestrogen resveratrol (R) is an analog of natural sex hormones and an estrogen receptor agonist [16]. R (3,4′,5-trihydroxystilbene) is a plant polyphenol occurring in grapes, berries, and nuts. It is also present in wines, first of all in red types. Plants produce R in response to stress, injury, or UV radiation and it helps the plant adapt to environmental conditions [17]. This compound has many properties, including activity against glycation, oxidative stress, inflammation, neurodegeneration, several types of cancer, and aging [18,19]. However, R bioavailability and pharmacokinetics are still problematic under oral supplementation. To improve these parameters, we used an experimental vaginal gel with R and hyaluronic acid (R-HA gel) and tested its effects in vivo experiments. HA is an exogenic glucosamine glycan analog. It is part of vaginal mucus, a physiological moisturizer, lubricant, and an important component of the protective barrier of the genital tract [12,13]. Together with excipients (lactic acid, etc.), they form an original composition that still has no analogs in the pharmaceutical market.

Therefore, the study aimed to assess the impact of vaginal R-HA gel on the reproductive system in ovariectomized rats and decipher the potential mechanisms of its action.

## 2. Results 

### 2.1. Weight and Tail Temperature Dynamics

Weight gain in females after ovarian removal is an indirect confirmation of compliance with the model of hypoestrogenemia, which is a manifestation of lipid metabolism, a characteristic demonstration of the menopausal syndrome [20]. This could play a crucial role in developing metabolic syndrome, type 2 diabetes, and obesity due to the imbalance of sex hormones in the blood during menopause [21].

It was shown that at 28-day administration, the therapeutic effect of the R-HA gel was manifested by a slowing of the rate of weight gain (Table 1). In contrast to untreated OVX group, in which body weight gain was 23%, in the R-HA group after 4 weeks of treatment, body weight gain was not more than 5.96%.

Evidence of the systemic therapeutic effect of the R-HA gel was a significantly less pronounced symptom of fever at the root of the tail (this corresponds to the menopausal symptom of hot flashes in women). In OVX rats, tail temperature increased by 10% compared with intact females, while under topical R-HA treatment, it increased by only 1.0% (Table 1).

### 2.2. Sex Hormones Dynamics

R-HA topical treatment for 4 weeks was associated with an insignificant increase of estradiol (35.6 ± 2.32 vs. 28.54 ± 1.6 pg/mL, *p* = 0.072) and progesterone (15.47 ± 1.72 vs. 10.64 ± 1.07 pg/mL, *p* = 0.064) levels as compared to OVX rats. On the other hand, hormone levels did not reach the value of intact rats (Figure 1). Although the R-HA treatment did not result in reaching the primary levels, the trend of elevation of sex hormones was detected. So, R-HA treatment provided a limited effect on systemic levels of estradiol and progesterone in OVX rats.

### 2.3. Atrophic Changes in Genital Tract after Ovariectomy

In the OVX rats, reductions up to three times of endometrium and approximately two times of vaginal mucosa thickness were found (Figure 2). Thinning of the endometrium was associated with the complex changes affecting both the luminal epithelium and glands. The epithelial lining was thinner with a higher nuclear–cytoplasmic ratio. Lamina propria demonstrated a dramatic decline in a number of glands. Only a few small glands were found in the endometrium of ovariectomized animals.

Vaginal mucosa in OVX group also demonstrated a decrease in thickness due to thinning of squamous stratified epithelium with altered cell differentiation. There were features of inflammatory infiltration of lamina propria by polymorphonuclear leucocytes (PMN) as compared with the control group (Figure 3). These changes were associated with the increased COX-2 expression in both stratified squamous epithelium and lamina propria infiltrated by highly immunopositive cells.

### 2.4. R-HA Gel Attenuates Endometrial Atrophic Changes and Inflammation in Ovariectomized Rats

R-HA gel administration attenuated the atrophy of the endometrium (382.79 ± 17.18 vs. 207.46 ± 10.7 µm, *p* < 0.001) and thinning of the vaginal epithelium (46.56 ± 1.76 vs. 32.33 ± 1.87 µm, *p* = 0.001) as compared with the OVX group, though the endometrial structure still differed from the control rats (Figure 2).

Endometrium demonstrated higher luminal epithelium and density of endometrial glands, which defined the increase of endometrial thickness. Glands were surrounded by numerous small vessels reflecting the ameliorating effect of R-HA administration. Assessment of vaginal mucosa revealed the anti-inflammatory effect of R-HA treatment as compared with OVX. This conclusion was confirmed by the decrease in COX-2 expression both in vaginal epithelium and stromal cells (Figure 3).

It was found that under R-HA gel treatment, the pathological manifestations of hypoestrogenism on the part of the vaginal mucosa were most clearly reduced. This correlated with a significant positive effect of the vaginal gel on the structural and functional state of the vaginal mucosa, as evidenced by histological studies, in particular reflected in an increase in epithelial thickness, the appearance of signs of typical estrogen-like action, and enhanced regenerative processes (pH 4.4–4.5) (Figure 4).

### 2.5. Resveratrol-Based Treatment Affects Bcl-2, Casp3, and VEGF Expression in the Endometrium

Deciphering the R-HA gel topical treatment mechanisms, we addressed the biomarkers involved in regulating apoptosis. While ovariectomy was associated with the enhanced Casp-3 and reduced Bcl-2 immunoreactivity in the endometrium, R-HA administration harmonized proapoptotic stimuli and increased Bcl-2 expression, providing an antiapoptotic effect in the endometrium.

Ovariectomy enhanced Casp-3 immunoreactivity in the lining epithelium, endometrial stroma, and glands of OVX animals (*p* < 0.001). R-HA administration was associated with a decrease of Casp-3 expression in all cell types, with the most profound effect in glands as compared with the OVX group (*p* < 0.001) (Figure 5).

While assessing the effect of R-HA on Bcl-2 expression, we found tissue-specific peculiarities. Normal endometrium demonstrated mild Bcl-2 expression in epithelium and gland and moderate to high expression in stromal cells. Ovariectomy impaired Bcl-2 expression mostly in glandular cells. At the same time, luminal epithelium demonstrated the changes in the pattern of Bcl-2 expression with the transition from mild diffuse pattern under normal conditions to intercalated distribution of cells with high and weak immunostaining in ovariectomized rats with no impact on total score (*p* = 0.561) (Figure 6). Although R-HA administration did not significantly change the score of Bcl-2 in luminal epithelium and stroma of endometrium, we found the most prominent alternative effect of R-HA treatment on Bcl-2 expression in glands as compared with OVX (*p* < 0.001), which would provide protection against apoptosis and endometrial atrophy.

Ovariectomy was associated with increased Casp-3 but decreased Bcl-2 expression, especially in endometrial glands cells. The antiapoptotic effect of R-HA administration was associated with increased VEGF expression.

The anti-apoptotic effect of R-HA was also associated with elevated VEGF expression in endometrial glands (*p* < 0.001) and stromal cells (*p* = 0.007). Similarly, enhanced expression of VEGF was defined in both stratified squamous non-keratinized epithelium and vaginal lamina propria (Figure 6).

Thus, R-HA treatment attenuates atrophic changes in the genital tract by antiapoptotic pathways in the endometrium and stimulating angiogenesis in the endometrium and vagina.

## 3. Discussion

Since menopause is not a disease but a manifestation of age-related changes, it leads to an endocrine imbalance in the body, which leads to the development of symptoms such as hot flashes, irritability, sleep disturbances, genitourinary disorders, and increases the risk of metabolic, cardiovascular–vascular, neurological diseases, and osteoporosis [22,23]. With the onset of natural menopause, estrogen levels decline over several years, and women’s bodies adapt to life in conditions of estrogen deficiency. During surgical menopause, 75–90% of women develop post-ovariectomy syndrome within a few days after surgery and the clinical picture of an increasing lack of sex hormones develops rapidly [24]. The above justifies the need for medical and social measures to protect health, and maintain efficiency and proper quality of life of women in the peri- and post-menopausal periods, and develop effective and safe drugs for personalized prevention and treatment of menopausal disorders.

Classical estrogen-replacement therapy is associated with increased cancer risk and other pathologies [1,2,3,8,10], so there is an urgent need for safer treatment approaches. Recently, a clinical investigation was performed comparing the therapeutic effectiveness of alternative vaginal drugs, such as promestriene, an estrogen agonist, and sodium hyaluronate (NaH), a nonhormonal, water-based agent. Low-potency estrogen agonist promestriene and nonhormonal vaginal NaH applications were found to be similarly effective in treating menopausal vaginal atrophy [25]. This year, a review was published about the connection between R and reproductive health [16]. The properties of phytoestrogen R have been described, along with its effects on embryogenesis and spermatogenesis, the most common pregnancy-related complications, umbilical blood vessels, and women’s reproductive health overall [16,26]. Three months of R peroral therapy improves endothelial function and diminishes blood pressure in ovariectomized rats [27]. The 24-month randomized, controlled, crossover trial showed that supplementation with phytoestrogen resveratrol could improve aspects of well-being, including chronic pain, a common complaint in post-menopausal women [28]. However, R bioavailability and pharmacokinetics are still problematic under oral supplementation [16]. To improve these parameters, we used an experimental vaginal gel with R. That is why this investigation studied the pharmacological effect of the vaginal gel with R and HA on local (genitourinary) and systemic manifestations of hypoestrogenism on the model of bilateral ovariectomy (surgical castration).

Our results suggest that 28 days of treatment with R-HA vaginal gel can normalize the typical menopause symptoms. Therapy efficacy was manifested by slowing weight gain in animals, lowering the vaginal pH, and decreasing the features of inflammatory infiltration and atrophy detected in OVX rats. At the same time, we did not reveal a significant increase in sex hormone levels. Addressing the question of possible effects of R-HA treatment at the systemic level, it is important to consider that R possesses various effects on estrogen–estrogen receptor signaling. It was shown that R, being a phytoestrogen, can increase the expression of native estrogen-regulated genes. Next, R can directly stimulate ER as an agonist, activating mitogen-activated protein kinase (MAPK) and further signaling cascades [29]. Finally, the genomic effect of R on ER transcription was shown, so that R can work as an activator and sensitized ER in OVX animals.

In addition to the ER-stimulating effect, R was shown to modulate basic biological processes in most cells. For example, R decreases cell proliferation, induces apoptosis in cancer cells, and stimulates smooth muscle relaxation [16,30,31]. This polyphenol also has anti-inflammatory, antioxidant properties, and cytoprotective effects [32,33,34]. The antioxidant effect of R relies on increasing the expression of various antioxidant enzymes and reducing superoxide production through up-regulating the tetrahydrobiopterin-synthesizing enzyme GTP cyclohydrolase I with the effect on endothelial nitric oxide synthase (eNOS). This provides a protective effect on the cardiovascular system [35]. Other favorable effects attributed to R are anti-lipid, anti-aging, anti-bacterial, anti-viral, and neuroprotective actions [35]. Polyphenols also have a slimming effect [35,36]. Besides the genomic effect of R was demonstrated to be realized through modulation of sirtuin 1 and the nuclear factor E2-related factor-2 expression with the following epigenetic regulation of various genes transcription [36]. In our study, we found a profound positive effect of local R-HA on vaginal mucosa health and structure. Low vaginal pH is due to lactic acid in the mucous [37]. In fact, lactic acid accumulation in vagina depends on both squamous epithelial cells maturation depending on sex hormones, and microbial flora. While ovariectomy resulted in vaginal pH increase and inflammatory infiltration of vaginal wall, R-HA treatment ameliorated vaginal homeostasis by reducing inflammation and decreasing pH level. Moreover, our data clearly demonstrated that local R-HA administration prevented atrophic changes in vaginal epithelium and provided an anti-inflammatory effect. The latter could be related to both normalizing vaginal microflora and the direct anti-inflammatory effect of R. The gel’s local activity can also be associated with the movement of hyaluronic acid. It was suggested that hyaluronic acid has a profile of efficacy, safety, and tolerability comparable with vaginal estrogens for treating symptoms of vaginal atrophy. It is a possible alternative for women who cannot use hormonal therapy [38].

For deciphering the mechanisms of the R-HA gel effect on endometrium and vagina, the immunohistochemical study was conducted using the following biomarkers: COX2, Casp-3, Bcl-2, and VEGF. Previous studies demonstrated that estrogens are essential for reproductive system cell survival and possess profound antioxidative and anti-inflammatory effects. Since endometrium and vagina are highly sensitive to sex hormone levels, a lack of estrogens resulted in a profound alteration of the balance between cell proliferation, survival, and apoptosis [39]. Naturally, OVX is accompanied by an elevation in levels of oxidized glutathione, lipid peroxidation, and mitochondrial DNA damage [39]. Such alterations can induce both oxidative stress and inflammation. Not surprisingly, OVX rats demonstrated a significant increase in the expression of Casp-3, which plays a key role in the execution phase of apoptosis. Alternatively, topical treatment with R-HA attenuated proapoptotic activation associated with the elevation of Bcl-2—mitochondrial protein contributing to antiapoptotic signals. In fact, R-HA administration reversed changes induced by ovariectomy. The beneficial effect of R-HA gel can be related to the impact of both resveratrol and hyaluronic acid. HA is abundant in mucosal connective tissue and contributes to moisture and nutrient transportation. On the other side, HA can support the proliferative phenotype of fibroblasts and protect them from apoptosis [39]. First, R positively impacts estrogen receptor alpha, stimulating their expression and signaling [40]. Second, resveratrol facilitates angiogenesis by enhancing VEGF production, which is essential for appropriate blood supply and preventing sclerotic changes [41]. Finally, harmonizing oxidative stress events, pro- and anti-apoptotic signaling are necessary for tissue homeostasis maintenance and reducing inflammation.

In this pilot study, the treatment with R-HA vaginal gels effectively improved women’s reproductive health. It is an excellent alternative, especially for patients who cannot use hormones. Therefore, further preclinical and clinical studies are needed to confirm the positive results of R-HA gel. Larger and randomized further studies must confirm these findings.

## 4. Materials and Methods

### 4.1. Ethics Statement

This study was carried out in strict accordance with the recommendations in the Guide for the Care and Use of Laboratory Animals of the National Institutes of Health and the general ethical principles of animal experiments (Strasbourg, 1985), approved by the First National Congress on Bioethics Ukraine (September 2001), and the Guide for the Care and Use of Laboratory Animals, approved by the bioethics commission of Bogomolets National Medical University (Protocol number: 4/2021).

### 4.2. Study Design

The study was carried out on 21 Wistar female rats aged 3–6 months. The animals were raised in the vivarium nursery of the Bogomolets National Medical University. Rats were kept in a clean, ventilated room with a controlled temperature (20–25 °C), relative humidity (50–55%), and a 12 h light–dark cycle. They were fed standard laboratory rodent chow with water available ad libitum. The rats were acclimatized for one week before the start of the experiment. Animals were divided into 3 groups (Table 2): (1) intact rats, (2) ovariectomized rats (OVX), and (3) OVX group treated with vaginal 0.5% R-HA gel (0.1 mL/day, daily 28 days, R-HA).

To model the hypoestrogenic state in animals, accompanied by symptoms similar to menopausal symptoms in women, removed the main hormone-producing gland—the ovaries. Wistar female rats were OVX as previously described [42]. Bilateral ovariectomy of females was performed in aseptic conditions under thiopental anesthesia (70 mg/kg, intraperitoneal). After castration, the females were kept in an accessible model for five weeks. This time was necessary for developing hypoestrogenemia. On the 35th day of the experiment, rats were administered, with the help of an insulin syringe with an atraumatic tip once a day for 28 days, 0.1 mL of test samples of vaginal gels.

### 4.3. Drug

Test samples of vaginal gel with R-HA were developed under the guidance of Professor Ruban O.A. at the Department of Industrial Technology of the National University of Pharmacy (Kharkov). The gel contains R, which is a polyphenol, a phytoestrogen that does not require prior metabolism to detect pharmacological action (unlike soy isoflavones), HA, and excipients, including lactic acid (LA). The R substance contained 50% trans-resveratrol of plant origin, obtained from *Polygonum Cuspidatum*. According to dose escalation studies, it was found that the optimal content of R in the dosage form is 0.5%, because of the vast majority of indicators in the model of experimental hypoestrogenic conditions in animals. This concentration received the highest scores [43].

### 4.4. Endpoints Assessment

The following endpoints were used for assessing the effect of R-HA gel effects in 28 days of the experiment: the body weight, tail temperature, vaginal pH, serum estrogen, and progesterone. Besides, a histological examination of the structure of the endometrium and vaginal mucosa was conducted. For deciphering the mechanisms of the R-HA gel effect on endometrium and vagina, the immunohistochemical study was conducted using the following biomarkers: cyclooxigenase 2 (COX2), caspase 3 (Casp-3), Bcl-2, and vascular endothelium growth factor (VEGF). Cyclooxigenase 2 is an inducible form of Prostaglandin-endoperoxide synthase involved in prostanoid biosynthesis during inflammation. As ovariectomy was shown to be associated with the increased inflammatory response to various stimuli [44]; the assessment of COX2 expression will allow the evaluation of the effect of R-HA-based treatment on inflammatory pathways in the female reproductive system. The expression of pro-apoptotic agent Caspase 3 (cysteine proteinase playing a central role in the execution phase of cell apoptosis) and Bcl-2 (B cell lymphoma/leukemia-2), an integral mitochondrial membrane protein blocking the apoptotic death of cells, was shown to be related to ovarian hormones levels [45]. Ovariectomy or hypoestrogenic conditions can disrupt cell survival, inducing apoptosis and endometrial atrophy. As R acts as an estrogen receptor (ER) agonist [40], R-HA treatment can modulate signaling pathways in the endometrium of ovariectomized rats. To test this hypothesis, we evaluated Casp-3 and Bcl-2 expressions in the endometrium of experimental animals. Finally, considering the vasoprotective and angiogenesis-promoting effects of R [46,47], we assumed that R-HA treatment could maintain angiogenesis. To check this hypothesis, we examined the expression of VEGF, regulating endothelial cells’ survival and proliferation during angiogenesis. 

The body weight of rats was measured on laboratory electronic scales FEH-300 (AnD, Ukraine) from 9:00 to 10:00 daily.

All animals were fasted for about 12 h before sacrifice, which was performed by cervical dislocation under urethane anesthesia. Blood was gathered into a microtube containing a mixture of EDTA and NaF in a 1:2 ratios. The content of hormones in the serum of female rats was performed by enzyme-linked immunosorbent assay on the enzyme-linked immunosorbent assay Stat Fax 303 plus (Awareness Technology, Palm City, FL, USA) using standard sets of reagents “Estradiol ELISA” and “Progesterone ELISA” (LLC “HEMA”, Kharkiv, Ukraine) according to instructions.

### 4.5. Histological Analysis and Immunohistochemical Markers Assessment

Uterus and vagina were taken under deep anesthesia, fixed in 10% neutral buffered formalin, and processed automatically. Paraffin sections, 4 µm thick, were cut from each sample, stained by hematoxylin and eosin according to the standard protocols, and observed using a Leica light microscope.

For IHC, serial sections 4 μm in thickness were used. Tissues were deparaffinized and hydrated. Endogenous peroxidase activity was blocked using 3% methanol in hydrogen peroxide. Next, antigen retrieval in a water bath at 98 °C was performed using Tris EDTA or citrate buffer (pH6) followed by primary antibodies incubation. After washing, labeled polymer secondary antibodies (Envision Detection System, Dako, CA, USA) were added to the slides. Peroxidase activity was detected using diaminobenzidine (DAB)—tetrahydrochloride liquid plus Chromogen System (Dako) substrate. The reaction was stopped with distilled water. After that, sections were counterstained with hematoxylin and mounted in Richard–Allan Scientific Mounting Medium (ThermoFisher, Waltham, MA, USA). The following antibodies were used: VEGF, Casp-3, Bcl-2, and COX-2.

Assessing the endometrium, its thickness was measured. Besides, the expression of pro- and antiapoptotic markers (Casp-3 and Bcl-2) as well as VEGF was evaluated in different cells including luminal epithelium, stromal cells, and endometrial glands.

Evaluating changes in vaginal mucosa included the measurement of its thickness, assessment of epithelial lining, and inflammatory reaction. The last was assessed using the semiquantitative score from 0 to 3 (0—no inflammatory reaction, 1—mild, 2—moderate, and 3—severe inflammatory reaction). In addition the expression of COX-2 and VEGF was considered.

Assessment of immunostaining for the markers mentioned above was based on using the combined semiquantitative immunohistochemical score that included both intensity score (0—no staining, 1—weak, 2—moderate and 3—strong) and its spread defined by the percentage of immunopositive cells. The percentage of positive cells was categorized as 0—negative (≤5%), 1—mild (6–25%), 2—moderate (26–75%), or 3—intensive (>75%). The total score of immunostaining was calculated as a multiplication of the intensity of staining and the percentage of immunoreactive cells. The assessments of immunohistochemical staining and morphometric measurement were performed by two independent observers.

### 4.6. Statistical Analysis

Statistical analysis performed by using SPSS-21 and GraphPad Prism 7 software. All data were expressed as mean ± standard error (M ± SEM). Data distribution was analyzed using the Kolmogorov–Smirnov normality test. Normally distributed continuous variables were analyzed with Analysis of Variance (one-way ANOVA) and if the results were significant, a post-hoc Turkey’s test was performed. Non-normally distributed continuous variables were assessed with non-parametric Kruskall–Wallis test. The changes of vaginal secretions pH participants in female rats before and after treatment were compared by paired sample *t*-tests. *p* value of <0.05 was considered statistically significant.

## 5. Conclusions

In this study, new scientific data were obtained to substantiate the feasibility of developing an innovative topical drug with R and HA. Twenty-eight-day application of this gel demonstrated the effectiveness of therapeutic action in an experimental model of hypoestrogenic conditions. It elucidated certain mechanisms of its anti-inflammation action. Namely, the present study revealed a novel COX-2-dependent anti-inflammation mechanism underlying the stimulated Bcl-2 expression, providing an antiapoptotic effect in the endometrium and vagina. This effect was associated with the stimulating effect of the treatment on angiogenesis through elevated VEGF expression in endometrial glands and stromal cells.

The limitations of the study that there are only a few experimental animals included and the absence of molecular biology methods such as Western blotting, etc. for results interpretation.

The obtained data proved the pharmacological effectiveness of using the R-HA vaginal gel to reduce the symptoms of menopause and constitute an experimental rationale for further preclinical and clinical research.

## Figures and Tables

**Figure 1 molecules-27-04916-f001:**
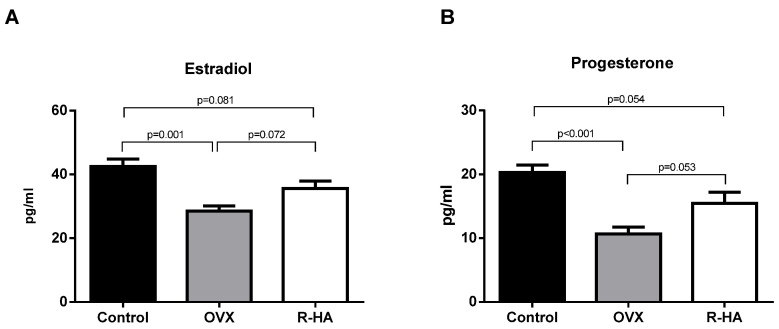
Serum sex hormones levels in control, OVX, and after 4 weeks of R-HA administration ((**A**)—estradiol; (**B**)—progesterone). Data are presented as the M ± SEM. One-way ANOVA with post hoc Tukey’s test for multiple comparisons was performed for data analysis.

**Figure 2 molecules-27-04916-f002:**
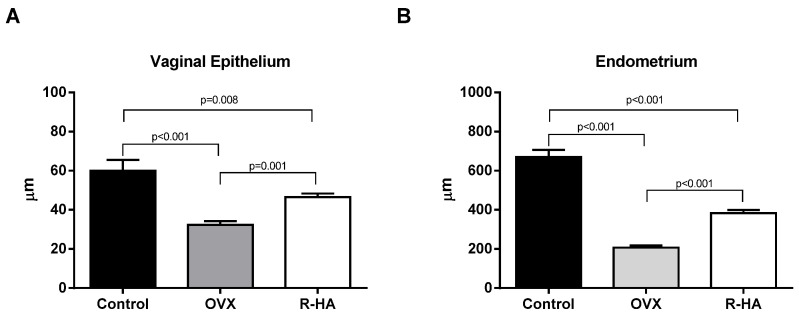
Effect of ovariectomy and R-HA gel administration on the thickness of vaginal epithelium (**A**) and endometrium (**B**). Data are presented as the M ± SEM. One-way ANOVA with post hoc Tukey’s test for multiple comparisons was performed for data analysis.

**Figure 3 molecules-27-04916-f003:**
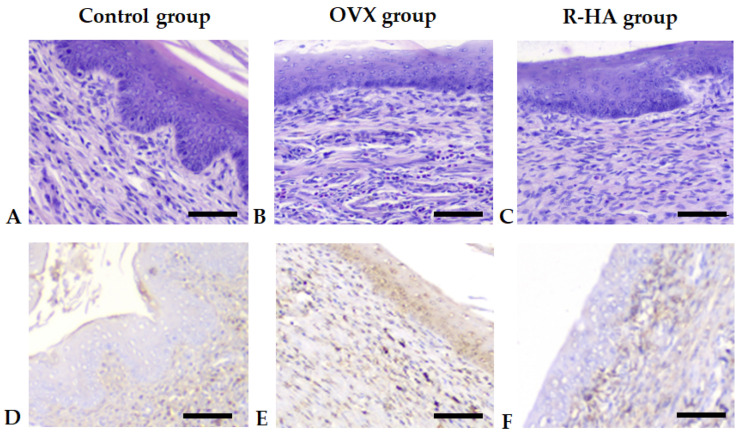
Impact of ovariectomy and R-HA treatment on vaginal mucosa. (**A**–**C**)—H&E staining. Scale bars 50 µm. (**D**–**F**)—Immunohistochemistry demonstrating COX-2 expression. Scale bars 50 µm. (**A**)—The vaginal mucosa of the control group. (**B**)—The vaginal mucosa of the rats after ovariectomy demonstrates thinning of the stratified squamous nonkeratinized epithelium and inflammatory infiltration of the lamina propria by polymorphonuclear leukocytes. (**C**)—R-HA treatment of ovariectomized rats attenuated inflammatory infiltration and prevented covering epithelium atrophy. (**D**)—A few COX-2 positive cells in vaginal mucosa of the control group animals. (**E**)—Increased number of COX-2 positive cells in the vaginal mucosa. (**F**)—Decline of COX-2 expression in vaginal mucosa after R-HA treatment.

**Figure 4 molecules-27-04916-f004:**
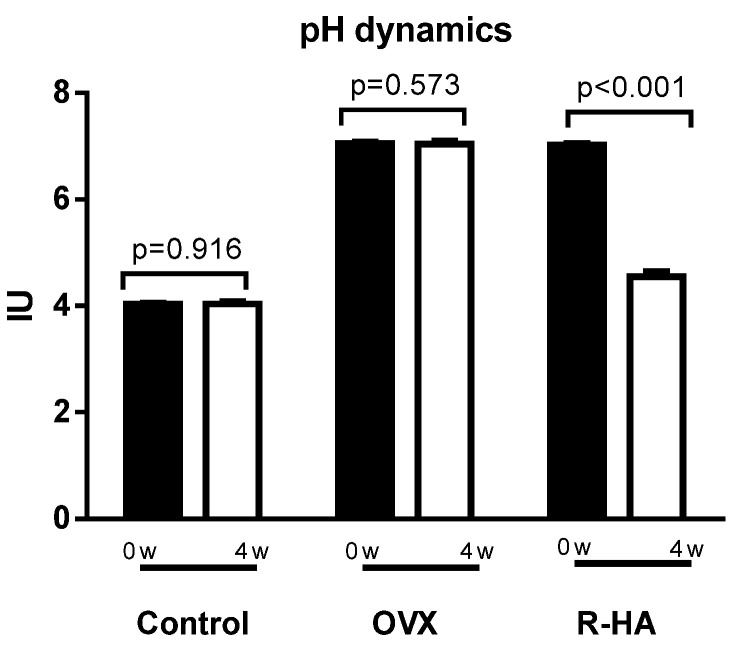
The value of the vaginal pH in control, OVX animals, and in female rats before and after R-HA topical treatment. Data expressed as mean ± SEM. For comparison, paired sample *t*-tests were used.

**Figure 5 molecules-27-04916-f005:**
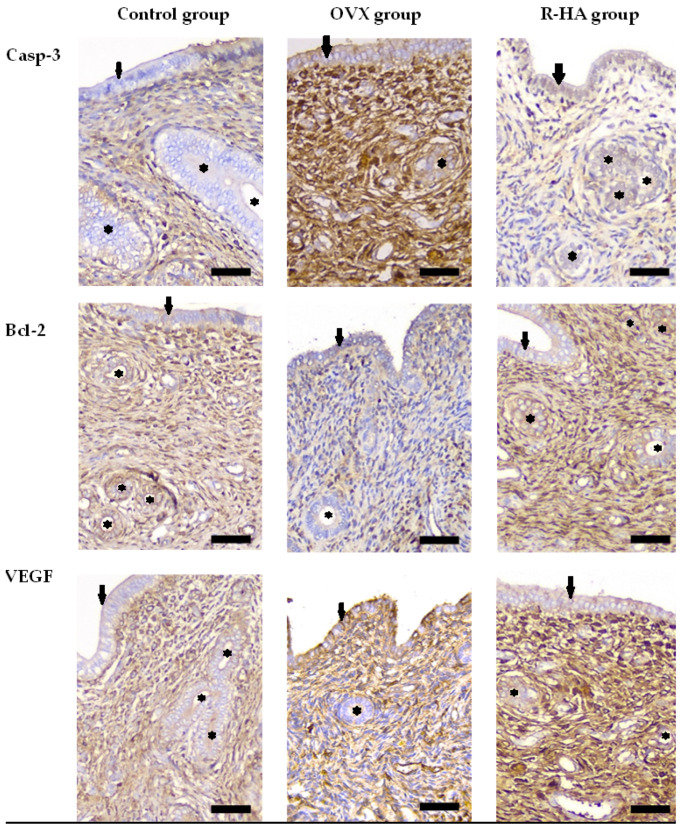
Changes in Casp-3, Bcl-2, and VEGF expression in experimental groups’ immunohistochemistry. Scale bars 50 µm. Ovariectomy resulted in thinning of endometrial lining epithelium (arrows), declining number of endometrial glands (asterisks), and changes in biomarkers expression. While Casp-3 expression increased in epithelium lining, endometrial stroma, and glands after ovariectomy, the levels of Bcl-2 dropped significantly in glandular cells as compared with the control group, shifting the balance between Casp-3/Bcal-2 towards pro-apoptogenic regulation. R-HA treatment attenuated the activation of pro-apoptogenic pathways through elevation and Bcl-2 expression and reduction of Casp-3 levels especially in glands. This effect was associated with enhancement of VEGF expression in both endometrial stroma and glands.

**Figure 6 molecules-27-04916-f006:**
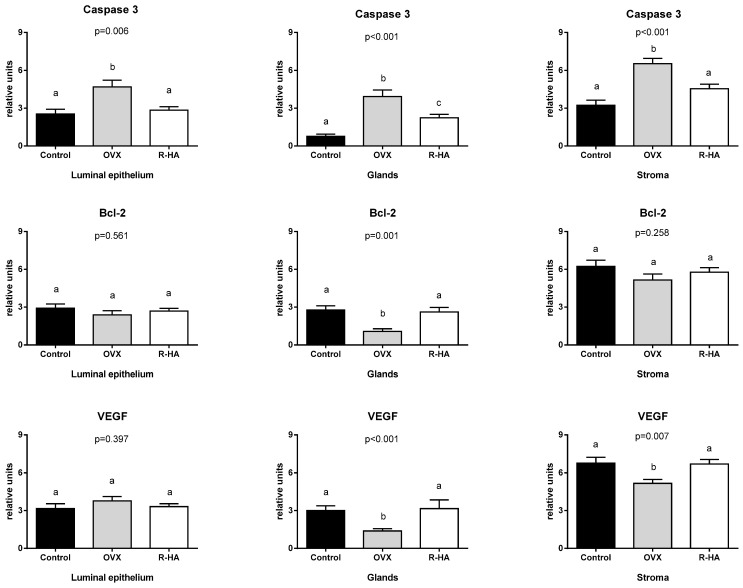
R-HA treatment attenuates proapoptotic changes in endometrium and recovers Bcl-2 and VEGF expression. Data are presented as the M ± SEM. One-way ANOVA with post hoc Tukey’s test for multiple comparisons was performed for data analysis. a, b, c values on the same row with different superscript letters show significant differences in *p* < 0.05.

**Table 1 molecules-27-04916-t001:** Weight and tail temperature dynamics before and after treatment.

Parameters	Intact Rats (*n* = 7)	OVX Group (*n* = 7)	R-HA Group (*n* = 7)
Weight before ovariectomy, g	208.67 ± 1.82 ^a^	211.33 ± 1.15 ^a^	218.00 ± 1.63 ^a^
Weight on week 5 after ovariectomy, g	214.17 ± 2.18 ^a^	246.33 ± 1.19 ^b^	245.33 ± 3.16 ^b^
Weight after 4 weeks of treatment, g	217.83 ± 2.51 ^a^	260.17 ± 1.92 ^b^	231.00 ± 2.46 ^c^
Tail t° before ovariectomy, °C	32.83 ± 0.14 ^a^	32.9 ± 0.12 ^a^	32.8 ± 0.8 ^a^
Tail t° on week 5 after ovariectomy, °C	32.92 ± 0.05 ^a^	36.18 ± 0.05 ^b^	35.67 ± 0.05 ^b^
Tail t° after 4 weeks of treatment, °C	32.94 ± 0.11 ^a^	36.15 ± 0.03 ^b^	33.12 ± 0.12 ^a^

Data are presented as the M ± SEM. One-way ANOVA with post hoc Tukey’s test for multiple comparisons were performed for data analysis. a, b, c Values at the same row with different superscript letters show significant differences at *p* < 0.05.

**Table 2 molecules-27-04916-t002:** Experimental groups and experimental conditions.

Experimental Group	Experimental Conditions	Dose According to the Drug Form
Intact rats (*n* = 7)	Intact animals	–
OVX (*n* = 7)	Females after bilateral ovariectomy	Water for injections, 0.1 mL/day
R-HA (*n* = 7)	Ovariectomized females were intravaginally administered 0.5% gel with R and HA	0.1 mL/day

## Data Availability

The data that support the findings of this study are available from the corresponding author upon reasonable request.

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
