# Peer review of "Resveratrol Effects on the Reproductive System in Ovariectomized Rats: Deciphering Possible Mechanisms"

_molecules, 2022, doi:10.3390/molecules27154916_

Round 1

Reviewer 1 Report

Manuscript 1819642 entitled “Resveratrol effects on the reproductive system in ovariectomized rats: deciphering possible mechanisms” of Ganna Zaychenko et al.

The research study explored the usefulness of vaginal gel containing resveratrol and hyaluronic acid for treating hypoestrogenic disorders. The study was conducted in vivo on Wistar female rats and the body weight, tail temperature, vaginal pH, estrogen and progesterone blood and immunohistochemical biomarkers (COX2, Casp-3, Bcl-2 and VEGF) levels were measured.

The study is relevant and carried out. Even if the results are interesting and well reported, the discussion of the project and the comment on the data obtained (discussion) must be reviewed and rewritten in a more punctual and integrated manner with further bibliographical references.

The manuscript can be accepted after a major revision of the discussion to the experimental data obtained.

pay close attention to the formatting of bibliographic references, follow the instructions of the newspaper. Standardize titles, add pages where they are missing, pay attention to the names of the authors, always use the capital letter 

Reviewer 2 Report

The study has been demonstrated the improvement in resveratrol (R) bioavailabilty and pharmacokinetics parameters along with hyaluronic acid (HA) in ovariectomized rats. A few minor comments need to be addresses:

Page 1; Line 20-21: It would be more appropriate to replace "these parameter" to "bioavailability and pharmacokinetics property".

Page 1; Line 27-28: The results in the abstract needs to be constructed in a way that all the outcomes with the significant figures should be given either in the percentage or statistically significant values.

Page 5 & 6; Figure 1 & 2: There is a need of comparison bar over the histogram in Figure 1 & 2 to show clearly the statistical significant differences similar to Figure 4.

The authors showed an interesting study results of Phytoestrogen resveratrol (R) and hyaluronic acid (HA) to assess the impact of vaginal R-HA gel on the reproductive system in ovariectomized rats. The abstract need minor attention as comments have been provided. The introduction has been written scientifically and professionally well, followed by methods description and results which were supported by statistical analysis, tables and figures. 

Reviewer 3 Report

Title: “Resveratrol effects on the reproductive system in ovariectomized rats: deciphering possible mechanisms

Authors: Ganna Zaychenko , Olena Stryga , Oksana Sinitsyna , Anna Doroshenko , Oksana Sulaieva , Tetyana Falalyeyeva , Nazarii Kobylia 

Comments:

Due to existing issues of the phytoestrogen resveratrol with regard to its bioavailability, the authors established an in vivo experimental model of ovariectomized rats to study the impact of a vaginal gel consisting of resveratrol and hyaluronic acid for optimizing pharmacokinetics and bioavailability. Results obtained by the authors comprised decreeing symptoms of hypoestrogenic disorders by using this composited gel, why they suggest further drug development using resveratrol and hyaluronic acid.

Major Points:

1) Grammar/English should be basically checked again.

2) Page 1 line 37: here it should be clarified that menopause can be ONE of the reasons for the diseases then described, but as here it sounds like it is the only reason

3) Page 2 line 85: here a little more information about the plant compound resveratrol would be nice

4) Page 2 line 90: here I would state in the experiments that in vivo experiments were done.

5) Page 4 line 143: brief explanation of why exactly these markers are being analyzed.

6) Page 5 line 215/216: what does this result say or imply? Short explanation

7) Page 7 Figure 3: A more detailed description would be helpful here: what can be seen? How can it be detected? What is to be observed? What does it imply? Small arrows or other markings could also be helpful with the individual images to make it clear what is important here and what to look for.

8) Page 8 Figure 5: nice pictures, but again a marker or accompanying explanation would be helpful.

9) Page 9 line 317: what do these results mean?

10) Page 9 Figure 6: the y-axis label "score" should be clarified: what score?

11) Page 9 line 323-326: explain meaning.

12) Page 9 line 327-329: justify statement

13) Page 10 line 350-356: Reference?

14) Page 11 line 390: explanation/definition of "Casp-3" needs to come earlier in the paper (has been used before, but not explained)

15) Page 11 Conclusion: the limitations of the study should also be discussed here: There are only a few experimental animals ("subjects"); also no molecular biology methods such as Western blotting, etc.

16) What are the perspectives?

Round 2

Reviewer 3 Report

The authors have satisfactorily addressed the concerns raised in the original version. The revised version is significantly improved. No further concerns.